# LEARNING SKILLS FROM ACTION-FREE VIDEOS

## ABSTRACT

Learning from videos holds great promise for enabling generalist robots by leveraging diverse visual data beyond traditional robot datasets. Videos often contain recurring skills (e.g., grasping, lifting) across different tasks and environments. While skill-based methods can acquire reusable behaviors, they typically rely on clean, action-labeled data, which limits their use in action-free video sources. On the other hand, existing learning-from-video methods often train monolithic models or focus on single-step dynamics, reducing their ability to extract and compose skills for efficient multitask learning and long-horizon planning. In this work, we introduce Skill Abstraction from Optical Flow (SOF), a framework for skill learning directly from action-free videos. To overcome the absence of action labels, we propose using optical flow as a surrogate for action and adapting the existing skill-learning algorithm to operate on flow-based representations. Our model learns to plan in the skill space and translates these flow-based plans into executable actions. Experiments show that our approach consistently improves performance in both multitask and long-horizon settings, demonstrating the ability to acquire and compose skills directly from raw visual data.

## 1 INTRODUCTION

Learning from videos offers a promising direction for scaling up data collection toward generalist robots (McCarthy et al., 2024). These large and diverse video datasets naturally capture the physical dynamics of the world and demonstrate how to complete tasks across a wide range of environments – capabilities that go beyond traditional robot data, which is often difficult to collect and lacks diversity. Prior work has explored leveraging such videos by learning video models Du et al. (2023); Ko et al. (2024); Bruce et al. (2024); Yang et al. (2024); Mendonca et al. (2023); Brooks et al. (2024); Zhou et al. (2024); Luo & Du (2025); Liang et al. (2024); Ajay et al. (2023b), reward models (Nair et al., 2022; Ma et al., 2023; Hung et al., 2025), representations (Srirama et al., 2024; Bahl et al., 2023), and latent actions (Ye et al., 2024; Schmidt & Jiang, 2024).

However, prior methods often learn from videos either at the trajectory level (Du et al., 2023) or at the single-step level (Ye et al., 2024), but both extremes miss what makes videos valuable beyond robot data. At the single-step level, videos provide only frame-to-frame supervision, which is too fine-grained and ambiguous without access to underlying dynamics – something better learned directly from real robot data. At the trajectory level, videos capture entire demonstrations, but this representation is overly specific, entangling all details of the scene and limiting the ability to transfer or recombine across tasks. We argue that the real benefit of videos lies in the middle ground: they expose reusable *action primitives* – structured skills such as grasping, pushing, or lifting – that humans naturally compose to perform diverse tasks as illustrated in Figure 1. For example, making tea involves reaching for a mug, grasping a kettle, and pouring hot water; the same primitives reappear in cleaning a table or watering plants. By identifying and composing these reusable primitives, a robot can learn modular skills that transfer to novel task compositions, thereby improving both sample efficiency and generalization ability.

Learning these action primitives (*skills*) over action sequences has been shown to benefit both multi-task learning and long-horizon planning in reinforcement learning (Pastor et al., 2009; Ajay et al., 2020; Pertsch et al., 2021; Nasiriany et al., 2022a; Shi et al., 2022; Laskin et al., 2022) and imitation learning (Zheng et al., 2024; Mete et al., 2024). However, these works typically rely on curated, action-labeled datasets that differ significantly from the nature of videos,

which are often noisy, unstructured, and lack explicit action labels. This work investigates a central question: How can we acquire transferable skills from action-free and diverse videos?

The core challenge lies in discovering abstractions that serve as surrogates for actions. Such abstractions should exhibit key properties of *skills*: they must be **similar across contexts** (e.g., "picking up an object" should generalize from a kitchen to a dining room), **composable**, enabling the construction of new tasks from existing motion patterns, and **representative of robot motion**, capturing the spatial and temporal structure necessary for generating physically executable trajectories.

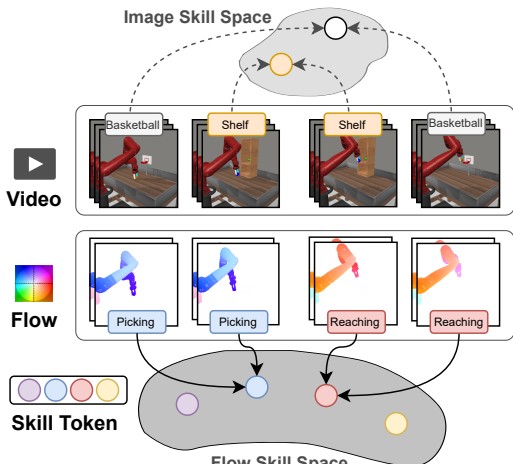

Figure 1: **Extracting skills from videos.** Videos contain composable *skills* that appear across different tasks and scenes. Learning and planning in a skill space enables efficient multi-task learning and long-horizon planning. Learning skills from raw images (*top*) often overfits to visual appearance. Instead, we learn skills from optical flow (*bottom*), which captures motion patterns and better reflects the underlying actions.

Learning directly from raw images (Janner et al., 2022; Ajay et al., 2023a) risks overfitting to scene-specific visual details or embodiment-specific cues. Alternatively, incorporating language (Du et al., 2023; Zhou et al., 2024) introduces a high-level semantic structure that can unify tasks from heterogeneous sources. Yet language is often ambiguous and underspecified – for instance, similar motions may be described in many different ways, making it less suitable for representing composable motion primitives. To sidestep these issues, we propose leveraging optical flow, which captures pixel-level displacements between successive frames. Optical flow has been shown to be a good mid-level dynamics representation for policy learning (Ko et al., 2024; Xu et al., 2024), yet its potential for skill learning remains underexplored.

In this paper, we introduce Skill Abstraction from Optical Flow (SOF), a framework for learning transferable skills directly from optical flow sequences. We begin by using off-the-shelf models to extract optical flow from raw videos. Next, we adapt existing skill extraction algorithms – originally designed for action sequences – to operate on optical flow. We then train a model to perform decision-making in the skill space to generate future flow plans. Finally, we map the high-level flow-based skill plan to low-level actions using a flexible module that can be either learned or learning-free. Our experiments on multi-task, long-horizon, and cross-embodiment settings demonstrate the effectiveness of skill abstractions.

We summarize our contributions as follows: (1) We propose SOF, a framework that learns skills from diverse, action-free videos by leveraging optical flow as an abstract representation. (2) Our experiments show that SOF effectively leverages skill abstraction and achieves strong performance on long-horizon tasks, opening new directions for skill discovery from action-free visual data.

## 2 RELATED WORK

**Learning robot policy from videos.** A growing body of work aims to leverage videos for robot learning without relying on expensive action-labeled datasets. One approach focuses on learning general visual representations from egocentric human videos (Grauman et al., 2022), which can then be applied to value function learning for downstream robot policy (Nair et al., 2022; Ma et al., 2023; Hung et al., 2025). Another line of work involves generative modeling, (Du et al., 2023; Ko et al., 2024) train video diffusion models to generate future states from in-domain videos, while others construct interactive simulators from large-scale internet videos (Bruce et al., 2024; Yang et al., 2024; Mendonca et al., 2023; Brooks et al., 2024; Zhou et al., 2024; Luo & Du, 2025; Liang et al., 2024; Ajay et al., 2023b). While these approaches are able to guide policies across tasks,

our approach takes this further by enabling planning over reusable skills, enhancing performance on long-horizon tasks.

Another line of work focuses on extracting actionable information from human videos. This includes modeling physical interactions (Jia et al., 2024; Agrawal et al., 2016), learning affordance-based representations (Srirama et al., 2024; Bahl et al., 2023), or leveraging latent action spaces (Ye et al., 2024; Schmidt & Jiang, 2024) from action-free videos. While (Ye et al., 2024; Schmidt & Jiang, 2024) are closely related to our work, our approach offers a complementary perspective by leveraging optical flow as a universal motion descriptor. This focus on motion patterns enables our work to capture shared dynamics.

**Learning from intermediate representations.** Beyond raw image observations, prior works have explored the use of intermediate representations like flow and keypoints for manipulation. (Wen et al., 2023; Bharadhwaj et al., 2024; Xu et al., 2024) predict future trajectories of object or robot arm keypoints to guide the policy. Recent works also show that optical flow is an effective representation for planning policies in dynamic environments (Ko et al., 2024; Gao et al., 2025). Despite promising results, these approaches primarily use intermediate representations as auxiliary guidance for policy learning. Whereas our method leverages optical flow as the core representation for skills extraction, this explicit use of flow for learning reusable motion primitives offers a more scalable framework for long-horizon robot planning.

**Skills for decision making.** To tackle long-horizon tasks, skill-based reinforcement learning (RL) (Pastor et al., 2009; Sutton et al., 1999; Schaal, 2006; Hausman et al., 2018; Nasiriany et al., 2022b; Zhang et al., 2022; 2024a) introduces temporal abstraction by representing policies as compositions of high-level skills or options (Sutton et al., 1999). A large body of prior work aims to discover such skills in an unsupervised manner to accelerate downstream tasks learning, typically using heuristics or contrastive learning to extract skills from offline data (Ajay et al., 2020; Pertsch et al., 2021; Nasiriany et al., 2022a; Shi et al., 2022; Laskin et al., 2022). Some approaches model low-level skills as discrete latent codes (Mete et al., 2024; Zheng et al., 2024). However, the aforementioned approaches require large amounts of action-labeled demonstration to learn meaningful skill representations. While most prior works rely on discovering skills with robot demonstrations, some recent works have explored learning skill representations from action-free videos (Zhu et al., 2022; Tomar et al., 2023; Xu et al., 2023), reducing the need for ground-truth state or action labels. However, these methods often assume access to structured video data, such as paired human-robot demonstrations or predefined skill boundaries. In contrast, our approach learns skill representations directly from raw videos without task supervision. By leveraging optical flow as a proxy for actions, our framework enables the discovery of temporally abstract, composable skills across diverse tasks and environments, offering a more scalable and general approach to skill-based decision making.

## 3 PRELIMINARY

### 3.1 PROBLEM SETTING

We consider an action-free video dataset denoted as $\mathcal{D}_{\text{video}} = \{(\boldsymbol{v}_i, \ell_i)\}_{i=1}^{M}$, consisting of $M$ video-language pairs. Each video $\boldsymbol{v}_i = (x_1, \ldots, x_T)$ is a sequence of RGB frames $x_t \in \mathbb{R}^{H \times W \times 3}$, and each corresponding language annotation $\ell_i$ is a natural language description of the task depicted in the video. In addition, we assume access to an action-labeled dataset, $\mathcal{D}_{\text{act}}$, which may either be a small subset of $\mathcal{D}_{\text{video}}$ with annotated actions or a larger dataset of interaction trajectories collected in the environment (e.g., play data). Notably, this dataset does not necessarily include language annotations.

### 3.2 LATENT VARIABLE MODELS FOR TEMPORAL ABSTRACTIONS

Latent variable models have been adopted to capture the temporal structure of decision-making problems in robotics (Jiang et al., 2023; Lee et al., 2024; Kong et al., 2024). Rather than modeling actions at the low-level resolution, these methods aim to learn compact latent spaces that summarize sequences of actions into higher-level abstractions.

We use the Quantized Skill Transformer (QueST) Mete et al. (2024) as our backbone model. QueST is a two-stage latent variable model that represents action sequences as sequences of discrete skill

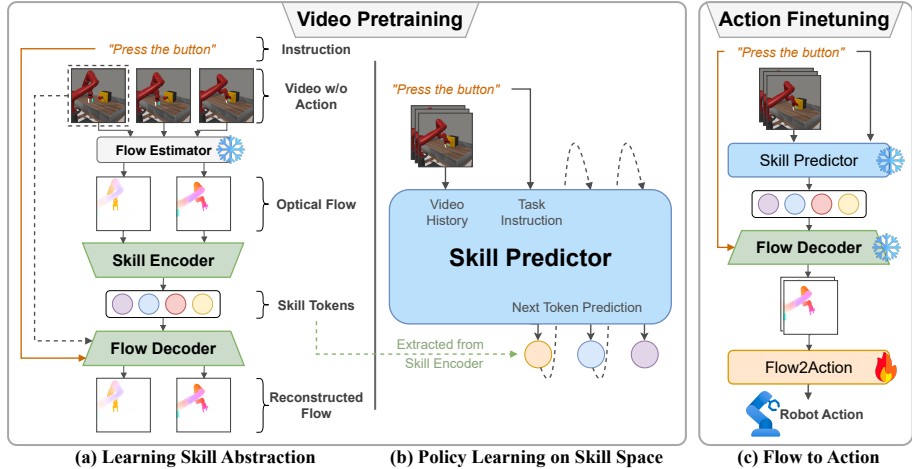

Figure 2: **Skill Abstraction from Optical Flow (SOF).** (a) Learn an action abstraction from optical flow using latent variable models to capture motion patterns across tasks. (b) Learn a skill predictor to perform policy learning in the skill space. (c) Given the first frame and an instruction, SOF generates a skill plan, decodes it into optical flow using a decoder, and infers actions using a lightweight Flow2Action module. The Flow2Action module can be either learned or calculated.

tokens. In the first stage, an autoencoder with causal strided convolutions and masked self-attention compresses fixed-length action sequences into multiple quantized embeddings using Finite Scalar Quantization (FSQ) Mentzer et al. (2024). In the second stage, a transformer-based policy models the distribution over skill token sequences conditioned on a window of observations and task embeddings.

## 4 METHODOLOGY

In this section, we present the key ideas and implementation details of SOF, Skill Abstraction from Optical Flow, as shown in Figure 2. Section 4.1 describes how we learn action abstractions that predict a sequence of skill tokens from a sequence of action-free videos. In Section 4.2, we introduce a skill policy that predicts these skill tokens based on video history and a given instruction, enabling policy learning in the skill space. The predicted skills are then decoded into optical flow conditioned on the current image. Finally, Section 4.3 outlines two approaches for inferring action sequences from the predicted optical flow: a learning-based method and a learning-free alternative.

### 4.1 LEARNING SKILL ABSTRACTIONS WITH OPTICAL FLOW

To enable the learning of *composable skill abstractions* from video datasets $\mathcal{D}_{\text{video}}$ composed of compound behaviors, our first step is to discover and aggregate recurring motion patterns into reusable skills across diverse demonstrations. However, with only videos, we lack both action labels and environment state transitions, making it challenging to infer actionable structure. To address this, we propose using optical flow as a surrogate for action labels. Optical flow offers several advantages: it is *action-directed*, capturing relative pixel movement that results from robot actions, and *noise-resistant*, as it focuses only on changes between consecutive frames while ignoring background and other motion-irrelevant noise.

Formally, given a video $\mathbf{v}_i = (x_1, \ldots, x_T)$, we compute optical flow between each consecutive frame using an off-the-shelf estimator, yielding a flow sequence $\boldsymbol{\delta}_i = \delta_1, \ldots, \delta_{T-1}$. In our implementation, we use FlowFormer++ (Shi et al., 2023) for real-world videos and NeuFlow-v2 (Zhang et al., 2024b) for simulated environments.

With the extracted flow sequences $\boldsymbol{\delta}_i$, we adapt the encoder-decoder architecture from QueST (Mete et al., 2024), which learns discrete skill abstractions from action sequences using a quantized autoen-

coder. In our setting, the encoder $\phi_\theta$ processes a flow segment $\delta_{t:t+H-1}$ of length $H$ and encodes them into a sequence of discrete latent skill tokens $\boldsymbol{c} = (c_1, \cdots, c_n)$ via Finite Scalar Quantization (FSQ):

$$\mathbf{c} = \text{FSQ}\left(\phi_\theta\left(\delta_{t:t+H-1}\right)\right) \tag{1}$$

The decoder $\psi_\theta$ reconstructs the original flow segment from these tokens. Crucially, we incorporate a positional inductive bias by conditioning the decoder on the initial frame $x_t$, leveraging the fact that optical flow captures relative motion. This conditioning enables the model to disentangle skill-relevant dynamics from absolute position, which may vary across demonstrations but is irrelevant to the underlying motion primitive.

The autoencoder is trained using a flow reconstruction loss:

$$\mathcal{L}_{\text{recon}}(\theta) = \|\psi_\theta(\text{FSQ}(\phi_\theta(\delta_{t:t+H-1})), x_t) - \delta_{t:t+H-1}\|_1. \tag{2}$$

## 4.2 LEARNING DECISION MAKING WITH SKILLS

After learning skill abstractions, we train a skill policy $\pi_\omega(c_t \mid x_t, e)$ to predict the skill based on the current frame $x_t$ and the task embedding $e$. The image observation is encoded using a learned vision encoder, which is trained jointly with the skill policy in an end-to-end manner. Notably, the model is also trained using the action-free video dataset $\mathcal{D}_{\text{video}}$.

Unlike prior methods such as ATM (Wen et al., 2023), which rely on multiple synchronized camera views, or QueST (Mete et al., 2024), which requires privileged state information, our method uses only third-person visual observations—similar to learning-from-video approaches (Du et al., 2023; Ko et al., 2024; Ye et al., 2024)—making it better suited for unstructured, real-world video data.

To model the temporal dependencies between skill tokens, we follow QueST (Mete et al., 2024) and employ a decoder-only Transformer as the skill predictor. The model autoregressively generates the skill sequence based on the current image and task context:

$$\pi_\omega(c_{1:n} \mid x_t, e) = \prod_{i=1}^{n} \pi_\omega(c_i \mid c_{<i}, x_t, e) \tag{3}$$

The skill policy is optimized using the negative log-likelihood objective:

$$\mathcal{L}_{\text{skill}}(\omega) = -\sum_{i=1}^{n} \log \pi_\omega(c_i \mid c_{<i}, x_t, e) \tag{4}$$

## 4.3 ACTION EXECUTION VIA PREDICTED OPTICAL FLOW PLAN

With the modules described in Section 4.1 and Section 4.2, our model predicts future optical flows that represent the actions a robot should take to complete a given task, based on the history of image observations and the task instruction. In this section, we introduce two approaches—one learning-free and one learning-based—to convert the predicted flows into executable actions.

**Learning-free** For the learning-free method, we adopt the action regression technique from AVDC (Ko et al., 2024), which infers actions directly from optical flow. Specifically, it estimates rigid transformations of the target object from the optical flow sequence, producing a series of SE(3) transformations. These transformations are then executed using a heuristic grasping strategy combined with a simple path-following policy to achieve the object transformation.

**Learning-based** However, AVDC relies on several assumptions about the environment, such as the availability of depth information and accurate segmentation masks, which may not be accessible or reliable in real-world scenarios. To overcome these limitations, we propose a learning-based alternative that frames flow-to-action mapping as a regression problem. We fine-tune a lightweight flow-to-action model using a small set of videos with ground-truth action labels, enabling the model to infer actions from flow in a data-efficient and environment-agnostic way. In our experiments, we show that this approach performs well even with limited labeled data, demonstrating strong potential for action inference compared to traditional methods such as inverse dynamics models (Agrawal et al., 2016).

# 5 EXPERIMENTS

Our experiments seek to answer the following questions: (1) Can SOF efficiently acquire diverse skills from multi-task video data? (2) Do the acquired skills enhance performance on long-horizon tasks? (3) Can the learned skills generalize across different embodiments?

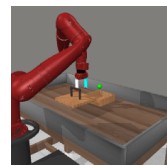 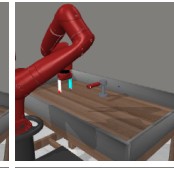 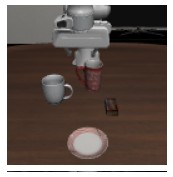 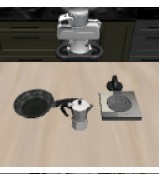 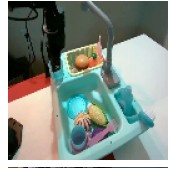 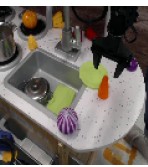

| (a) MetaWorld | (b) LIBERO | (c) BridgeData V2 |

Figure 3: **Environmental setups**: (a) **MetaWorld** is a simulation benchmark featuring a variety of manipulation tasks. We used it to evaluate multi-task performance and cross-embodiment generalization. (b) **LIBERO** is a simulation benchmark for lifelong robot learning. We use it to study multi-task and long-horizon performance. (c) **BridgeData V2** is a real-world dataset of manipulation behaviors. We use it to evaluate our skills on diverse environments and tasks.

## 5.1 BASELINES

**Behavior Cloning (BC).** We implement multi-task BC. Specifically, we concatenate image features with task instruction features encoded by CLIP, and feed the combined representation into a 3-layer MLP to predict actions. The model is trained using mean squared error loss.

**Diffusion Policy (DP)** (Chi et al., 2023) is a state-of-the-art imitation learning algorithm. We adopt the CNN-based Diffusion Policy, which uses a 1D convolutional U-Net to denoise action sequences sampled from a Gaussian prior, conditioned on RGB observations.

**AVDC** (Ko et al., 2024) uses a text-conditioned video diffusion model with a learning-free approach that infers actions from optical flow, depth, and object masks.

**LAPA** (Ye et al., 2024) is a vision-language-action model that learns from videos. It consists of a latent action pretraining stage on action-free video data $\mathcal{D}_{\text{video}}$, where the goal is to learn latent actions that can predict future frames. This is followed by an action finetuning stage on action-labeled data $\mathcal{D}_{\text{act}}$. Following the original setup, we use the 7B LWM-Chat-1M (Liu et al., 2025) as the base VLM.

## 5.2 ENVIRONMENTS

We evaluate on MetaWorld and LIBERO benchmarks. For multi-task settings, we use 9 MetaWorld tasks (third-person camera) and 10 LIBERO-GOAL tasks (front-facing camera). Each task includes 50 action-free videos ($\mathcal{D}_{\text{video}}$) and 10 action-labeled trajectories ($\mathcal{D}_{\text{act}}$), used for fine-tuning (LAPA) and Flow2Action (Ours). For long-horizon tasks, we select 4 from LIBERO-10, as most baselines fail on the rest with video-only supervision. Each has 50 action-free videos and 10 action-labeled trajectories. For cross-embodiment evaluation, we use MetaWorld to test generalization between Sawyer and Panda. See Section 5.5 for details. We also conduct a skill space analysis on BridgeData V2 (Walke et al., 2023) to test real-world skill abstractions. We learn a latent variable model for skill representation and train a skill predictor to infer flow plans.

## 5.3 MULTI-TASK LEARNING

We evaluate multi-task learning on the MetaWorld and LIBERO-GOAL benchmarks. On Meta-World, SOF consistently outperforms both video-based baselines and multi-task BC baselines, using the same amount of action-labeled data. We found that LAPA, which learns latent actions from action-labeled data, fails to perform grasping tasks effectively. On LIBERO-GOAL, which comprises tasks involving similar motions (e.g., picking and placing objects), our method successfully captures reusable skills and outperforms the baselines. However, we observe two limitations: it

Table 1: **Multi-Task Learning on MetaWorld.** SOF effectively utilizes action-free video data and outperforms all baselines. We compare against (1) multi-task BC baselines trained on action-labeled data $\mathcal{D}_{act}$ (10 demos per task), and (2) video-based methods trained on 50 demos per task that fine-tune or learn a module (e.g., IDM, Flow2Action) using $\mathcal{D}_{act}$. The highest score is highlighted in **bold**, and the second-highest score is underlined.

| | door-open | door-close | bin-picking | box-close | drawer-open |
|---|---|---|---|---|---|
| BC | $0.64 \pm 0.06$ | $\mathbf{1.00} \pm \mathbf{0.00}$ | $0.00 \pm 0.00$ | $\mathbf{0.20} \pm \mathbf{0.07}$ | $0.63 \pm 0.02$ |
| DP | $0.00 \pm 0.00$ | $0.84 \pm 0.05$ | $0.00 \pm 0.00$ | $0.00 \pm 0.00$ | $\mathbf{1.00} \pm \mathbf{0.00}$ |
| AVDC | $\underline{0.84} \pm 0.04$ | $\underline{0.92} \pm 0.04$ | $0.00 \pm 0.00$ | $0.04 \pm 0.00$ | $0.02 \pm 0.02$ |
| LAPA | $0.00 \pm 0.00$ | $0.00 \pm 0.00$ | $0.00 \pm 0.00$ | $0.00 \pm 0.00$ | $0.00 \pm 0.00$ |
| SOF (Ours) | $\mathbf{0.98} \pm \mathbf{0.03}$ | $\mathbf{1.00} \pm \mathbf{0.00}$ | $\mathbf{0.24} \pm \mathbf{0.07}$ | $\underline{0.12} \pm 0.07$ | $\underline{0.78} \pm 0.04$ |

| | faucet-close | faucet-open | handle-press | assembly | **Overall** |
|---|---|---|---|---|---|
| BC | $\mathbf{0.78} \pm \mathbf{0.04}$ | $\mathbf{1.00} \pm \mathbf{0.00}$ | $\mathbf{0.87} \pm \mathbf{0.03}$ | $0.00 \pm 0.00$ | $\underline{0.57} \pm 0.01$ |
| DP | $0.06 \pm 0.02$ | $0.86 \pm 0.07$ | $0.00 \pm 0.00$ | $0.00 \pm 0.00$ | $0.31 \pm 0.01$ |
| AVDC | $0.24 \pm 0.04$ | $0.78 \pm 0.02$ | $\underline{0.72} \pm 0.04$ | $0.00 \pm 0.00$ | $0.42 \pm 0.02$ |
| LAPA | $0.17 \pm 0.08$ | $0.28 \pm 0.11$ | $0.65 \pm 0.11$ | $\underline{0.12} \pm 0.04$ | $0.14 \pm 0.02$ |
| SOF (Ours) | $\underline{0.62} \pm 0.06$ | $\underline{0.99} \pm 0.02$ | $0.69 \pm 0.06$ | $\mathbf{0.82} \pm \mathbf{0.07}$ | $\mathbf{0.69} \pm \mathbf{0.02}$ |

Table 2: **Multi-Task Learning on LIBERO-GOAL.** SOF clearly outperforms the baselines on repetitive tasks – primarily those involving picking up objects and placing them elsewhere – highlighting the advantages of reusable skills. However, SOF underperforms compared to BC and DP on tasks that involve handling small objects (e.g., bottle) or require distinct motions (e.g., open a drawer).

| | put-bowl-stove | put-bowl-cabinet | push-plate-stove | put-bottle-cabinet | put-cream-bowl |
|---|---|---|---|---|---|
| BC | $\underline{0.38} \pm 0.23$ | $\underline{0.06} \pm 0.01$ | $\underline{0.31} \pm 0.10$ | $\mathbf{0.39} \pm \mathbf{0.05}$ | $\mathbf{0.16} \pm \mathbf{0.06}$ |
| DP | $0.10 \pm 0.05$ | $0.04 \pm 0.04$ | $0.18 \pm 0.04$ | $0.01 \pm 0.01$ | $0.02 \pm 0.03$ |
| LAPA | $0.00 \pm 0.00$ | $0.00 \pm 0.00$ | $0.00 \pm 0.00$ | $0.00 \pm 0.00$ | $0.00 \pm 0.00$ |
| SOF (Ours) | $\mathbf{0.64} \pm \mathbf{0.06}$ | $\mathbf{0.30} \pm \mathbf{0.06}$ | $\mathbf{0.57} \pm \mathbf{0.06}$ | $\underline{0.01} \pm 0.01$ | $\underline{0.07} \pm 0.02$ |

| | turn-on-stove | put-bowl-plate | put-bottle-rack | open-middle-drawer | open-top-drawer | Avg. |
|---|---|---|---|---|---|---|
| BC | $\underline{0.77} \pm 0.05$ | $\underline{0.14} \pm 0.09$ | $\underline{0.03} \pm 0.03$ | $\underline{0.06} \pm 0.05$ | $0.00 \pm 0.00$ | $\underline{0.23} \pm 0.06$ |
| DP | $\mathbf{0.94} \pm \mathbf{0.04}$ | $0.01 \pm 0.01$ | $0.00 \pm 0.00$ | $\mathbf{0.46} \pm \mathbf{0.13}$ | $0.00 \pm 0.00$ | $0.18 \pm 0.01$ |
| LAPA | $0.00 \pm 0.00$ | $0.00 \pm 0.00$ | $0.00 \pm 0.00$ | $0.00 \pm 0.00$ | $0.00 \pm 0.00$ | $0.00 \pm 0.00$ |
| SOF (Ours) | $0.71 \pm 0.06$ | $\mathbf{0.21} \pm \mathbf{0.07}$ | $\mathbf{0.03} \pm \mathbf{0.01}$ | $0.00 \pm 0.00$ | $0.00 \pm 0.00$ | $\mathbf{0.25} \pm \mathbf{0.02}$ |

struggles to detect small objects such as cream containers or bottles, and it fails to execute motions requiring large rotations, such as opening drawers. In the LIBERO environment, where scenes across tasks are visually similar and the gripper is not easily distinguishable, these challenges lead to frequent failures in image based method such as LAPA.

## 5.4 LONG-HORIZON TASKS

We compare SOF against BC and DP on LIBERO-10 for long-horizon tasks. As shown in Table 3, when using 10 demonstrations across all methods, both BC and DP struggle with long-horizon tasks, while our method effectively leverages reusable skills, plans in the skill space, and successfully completes long-horizon tasks. When the number of demonstrations for BC and DP is increased, their performance becomes comparable to ours.

Table 3: **Long-horizon on LIBERO-10.** Using 10 action-labeled demonstrations per task for BC, DP, and SOF, we observe that BC and DP struggle with long-horizon tasks, whereas SOF efficiently composes reusable skills to solve them. When the number of demonstrations for BC and DP is increased to 30, their performance becomes comparable to SOF.

| | put-soup_sauce-basket | turn-on-stove-put-moka-pot | put-mug-left-right | put-mug-left-pudding-right | Overall |
|---|---|---|---|---|---|
| BC (10 demos) | $0.00 \pm 0.00$ | $0.09 \pm 0.05$ | $0.00 \pm 0.00$ | $0.02 \pm 0.02$ | $0.03 \pm 0.01$ |
| BC (30 demos) | $0.00 \pm 0.00$ | $0.23 \pm 0.10$ | $0.03 \pm 0.00$ | $0.02 \pm 0.02$ | $0.07 \pm 0.03$ |
| DP (10 demos) | $0.00 \pm 0.00$ | $0.06 \pm 0.04$ | $0.00 \pm 0.00$ | $0.00 \pm 0.00$ | $0.01 \pm 0.01$ |
| DP (30 demos) | $0.02 \pm 0.03$ | $0.44 \pm 0.06$ | $0.09 \pm 0.02$ | $0.06 \pm 0.04$ | $0.15 \pm 0.01$ |
| SOF (Ours) | $0.08 \pm 0.03$ | $0.42 \pm 0.06$ | $0.06 \pm 0.01$ | $0.10 \pm 0.03$ | $0.16 \pm 0.01$ |

## 5.5 CROSS-EMBODIMENT TRANSFER

**Skill transfer for decision making.** We test whether learned skills generalize across robot embodiments using the Franka Panda and Sawyer arms. In stage one, videos from both arms are used to learn a shared skill space; in stage two, policy learning is trained on only one arm's data. Flow2Action modules are trained separately per arm to map optical flow to low-level actions. We define "topline" as performance when training on all tasks per embodiment, serving as the upper bound.

As shown in Table 4a, decision-making transfers well to unseen embodiments. Figure 6 shows both arms follow similar skill-token sequences for the same task, with only minor timing differences due to embodiment-specific dynamics. This indicates the shared skill space captures consistent high-level behaviors while accommodating low-level variations.

**Transfer for visual perception and decision making on unseen tasks.** We further test unseen task generalization by splitting 10 Metaworld tasks between the Panda and Sawyer, training both skill abstraction and decision making only on the assigned tasks (Appendix B.3). The results in Table 4b indicate that, despite the absence of training data for certain tasks for a robot, SOF can still use skills from other embodiments to achieve high performance.

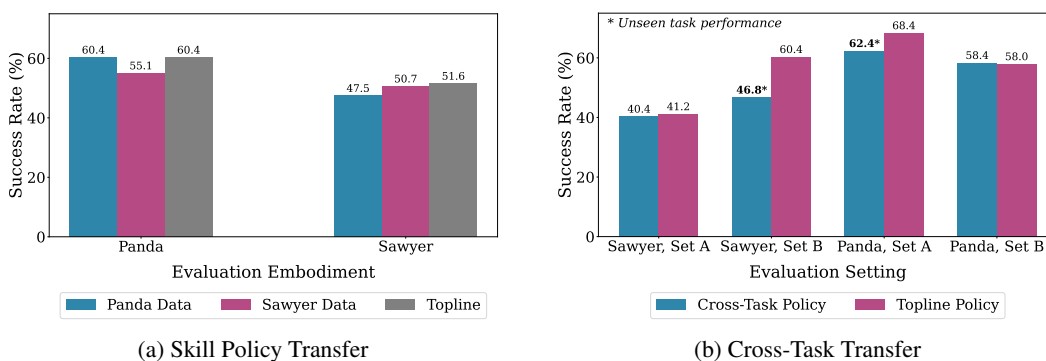

(a) Skill Policy Transfer  (b) Cross-Task Transfer

Figure 4: **Cross-embodiment transfer.** Average success rates on MetaWorld: **(a) Skill policy transfer.** We train the skill abstraction with both Panda and Sawyer data. In the policy training stage, only one embodiment's data is used. *Topline* shows results trained on all tasks per embodiment. The results show that shared skill representation enable effective transfer across embodiments. **(b) Cross-task transfer.** We partition tasks into disjoint sets, A and B. The *cross-task policy* is trained on Sawyer data from A and Panda data from B, while the *topline policy* is trained on the full dataset. The results indicate successful transfer, even when a task is unseen for one embodiment.

## 5.6 ANALYSIS

**Skill token analysis in multi-task setting.** We analyze learned skills in a multi-task setting to test whether similar motions map to the same skill across conditions. To handle the large codebook (1024), we cluster embeddings into 16 groups with K-means, then sample and visualize skills using the current frame and predicted optical flow for the next $k$ steps. The flow plan is illustrated with arrows: different colors indicate different directions, while color intensity reflects motion magnitude.

Figure 5 shows that tokens generalize across varied objects, tasks, and layouts: pushing and grasping share the same token (Fig. 5a), spatial invariance holds across positions and objects (Fig. 5b), and Bridge results (Fig. 5c) demonstrate consistent motion clustering despite real-world variability. These results highlight robust skill abstraction across simulation and real settings.

**Using optical flow vs. next-frame prediction as action surrogates for skill learning.** To verify the effectiveness of flow as an action representation, we investigate the use of next-frame prediction as an alternative to optical flow for skill learning. Specifically, during the skill learning phase, we replace the optical flow targets $f_1, \cdots, f_{T-1}$ with the sequence of future frames $o_2, \cdots, o_T$. This idea is inspired by LAPA Ye et al. (2024), which learns latent actions that induce the next frame. Accordingly, we substitute the Flow2Action model with an inverse dynamics model that predicts

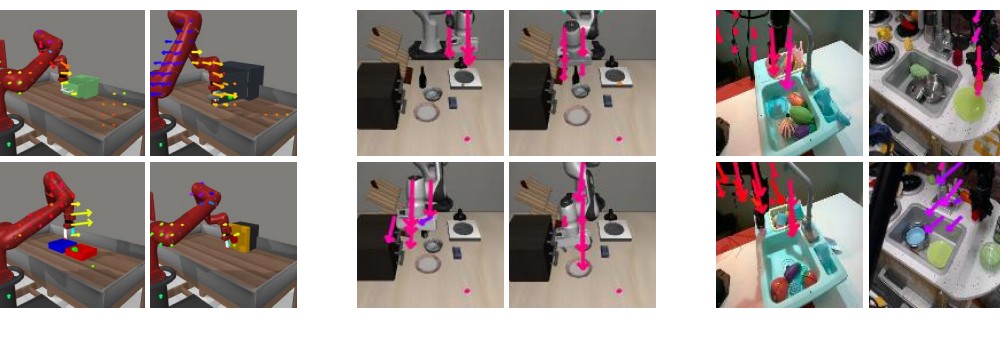

(a) Different Tasks & Scenes      (b) Different Objects & Positions      (c) Real World

Figure 5: **Skill Token Analysis**: Each figure shows the optical flow plan that corresponds to *the same skill token*. (a) Different tasks and scenes involving similar motion patterns are grouped together (b) Visually distinct objects in the same scene, positioned differently, are grouped together due to shared motion (c) Visually diverse real-world scenes are grouped together by shared motion patterns.

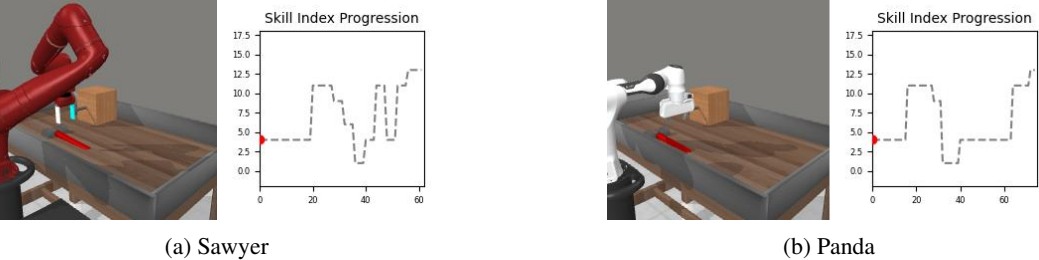

(a) Sawyer      (b) Panda

Figure 6: **Skill token index progress across embodiments.** Sawyer and Panda arms follow similar skill token sequences, with minor differences in skill duration.

actions based on the current and next frames. In our experiments, using next-frame prediction as an action surrogate results in a 13% lower success rate on MetaWorld compared to SOF.

We also analyze failure cases. As shown in Figure 7 in the Appendix, the model occasionally produces contradictory motions. For example, in the open faucet task, the agent initially moves toward the faucet but then reverses direction, suggesting it may have confused the "open faucet" skill with "close faucet" due to visually similar cues. A similar issue arises in the close door task: the agent first approaches the door as if to close it but then performs a motion in the opposite direction, resembling an attempt to open the door. These cases highlight the advantages of learning in flow space, where the model focuses on motion rather than appearance.

## 6 CONCLUSION

We introduced SOF, a framework for learning composable and transferable robotic skills directly from action-free videos by leveraging optical flow as a surrogate for action. SOF extracts structured motion primitives from raw videos, enabling policy learning in the learned skill space and translating these plans into executable actions via both learning-based and learning-free modules. Our experiments across multi-task, long-horizon, and cross-embodiment settings demonstrate that SOF improves performance over prior learning-from-video methods while requiring only minimal action supervision. The results highlight the potential of mid-level motion representations for scalable robot learning and open new directions for skill discovery from unstructured visual data.

**Limitation and future work.** Future work may extend SOF to broader data sources, such as human and egocentric videos, enabling more scalable and diverse skill learning beyond robot videos. In addition, our reliance on flow introduces certain limitations, such as occlusions between the robot arm and objects, sensitivity to visual instability, and dependence on fixed camera positions. To address these challenges, future work may explore alternative representations as action surrogates, such as extending to 3D using scene flow. We also aim to deploy our method in real-world settings to assess its practical applicability.

## 7 ETHICS STATEMENT

Our work focuses on improving the scalability and generalization of robot learning from unlabeled videos, which can benefit applications such as assistive robotics, home automation, and industrial manipulation. Since our method builds on publicly available datasets and models, and does not involve human subjects or sensitive data, we do not foresee any obvious negative societal impacts. Nonetheless, we encourage responsible use and emphasize that our framework should be applied in alignment with safety and ethical guidelines.

## 8 REPRODUCIBILITY STATEMENT

We have included the implementation details, training setup, training time, and hardware specifications in Appendix B and C to ensure reproducibility.

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

APPENDIX

# Table of Contents

## A  ADDITIONAL EXPERIMENTS

### A.1  LEARNING VS. LEARNING-FREE.

Table 4: **Learning-based *vs.* Learning-free**

|                 | door-close              | drawer-close            | faucet-open             |
|-----------------|-------------------------|-------------------------|-------------------------|
| Learning-based  | $1.00_{\pm 0.00}$       | $0.71_{\pm 0.04}$       | $0.99_{\pm 0.02}$       |
| Learning-free   | $1.00_{\pm 0.00}$       | $0.76_{\pm 0.02}$       | $1.00_{\pm 0.00}$       |

The Flow2Action module can be implemented either by training an end-to-end model that maps optical flow to actions or by using learning-free methods such as AVDC (Ko et al., 2024). In Table 4, we compare the performance of both approaches in MetaWorld. The results show that AVDC consistently outperforms the learning-based method. However, AVDC requires additional inputs (e.g., depth, segmentation) and prior knowledge about the environment to implement correctly. In our main experiments, we adopt the learning-based approach. Nonetheless, our Flow2Action module remains flexible, as it can be instantiated with either a learned model or a learning-free method depending on the application scenario.

### A.2  EXAMPLES OF FAILURE CASES IN NEXT-FRAME SKILL LEARNING

As shown in Figure 7, the model occasionally produces contradictory motions. For example, in the open faucet task, the agent initially moves toward the faucet but then reverses direction, suggesting it may have confused the "open faucet" skill with "close faucet" due to visually similar cues. A similar issue arises in the close door task: the agent first approaches the door as if to close it but then performs a motion in the opposite direction, resembling an attempt to open the door. These cases highlight the advantages of learning in flow space, where the model focuses on motion rather than appearance.

### A.3  FLOW ESTIMATOR ABLATION

To evaluate the robustness of SOF to different optical flow estimators, we compared RAFT-small (Eslami et al., 2024), GMFlow (Xu et al., 2022), and the more recent NeuFlow-v2 Zhang et al. (2024b). Table 5 reports the average success rates on MetaWorld tasks. While performance

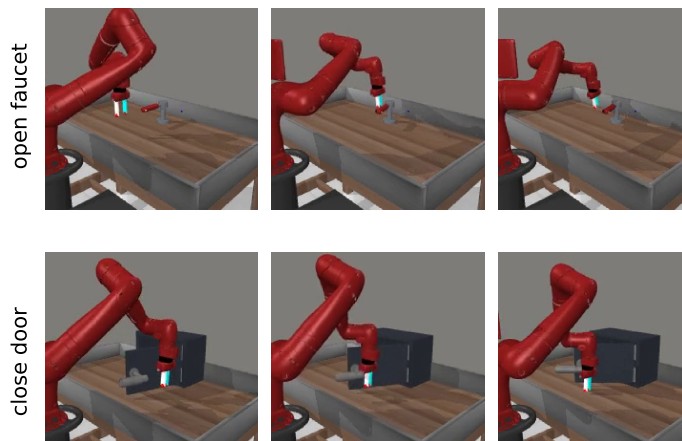

Figure 7: **Fail cases of next-frame skill learning**

degrades with weaker flow models, our framework remains competitive. In particular, GMFlow achieves results close to NeuFlow-v2 despite exhibiting slight background noise, indicating that SOF is resilient to moderate estimation errors. In contrast, RAFT-small shows a noticeable drop in success, likely due to its limited capacity, yet it still surpasses baseline methods. These results demonstrate that SOF is not overly reliant on a specific flow estimator and can adapt across multiple choices.

Table 5: **Flow estimator ablation on MetaWorld.**

| Flow estimator | Success Rate |
|---|---|
| RAFT-small | 0.61 |
| GMFlow | 0.67 |
| NeuFlow-v2 | 0.69 |

## A.4 DIRECT SKILL-TO-ACTION VS. FLOW-BASED DECODING

An alternative to our design is to bypass optical flow decoding and directly map discrete skill tokens into low-level actions. We implemented two such variants: (i) a fully-connected (FC) head and (ii) a transformer decoder conditioned on the input image. Table 6 summarizes the results. Both direct mapping approaches underperform significantly, with average success rates of 0.15 and 0.21, compared to 0.49 achieved by our flow-based Flow2Action module. Although direct mapping is computationally lighter, the intermediate flow representation provides a strong motion-centric prior that guides action inference, improving generalization across tasks. This suggests that optical flow serves as a valuable structured intermediate signal, and the slight overhead introduced by decoding into flow is a worthwhile trade-off for substantially higher performance.

Table 6: **Direct skill-to-action vs. flow-based decoding.**

| Method | Avg. Success Rate |
|---|---|
| Skill2Action (FC) | 0.15 |
| Skill2Action (Transformer) | 0.21 |
| Flow2Action (ours) | 0.49 |

# B EXPERIMENT DETAILS

## B.1 TRAINING HYPERPARAMETERS

The hyperparameters of all training stages of SOF are listed in Table 7.

Table 7: **Training hyperparameters.**

| Stage | Hyperparameter | value |
|---|---|---|
| Stage 1 | Encoder Dim. | 256 |
| | Eecoder Dim. | 256 |
| | Skill block size | 32 |
| | Downsample factor | 4 |
| | Attn. Dropout | 0.1 |
| | Encoder heads | 4 |
| | Encoder layers | 2 |
| | Decoder heads | 4 |
| | Decoder layers | 4 |
| | VQ type | fsq |
| | Codebook Size | 1024 |
| | Learning Rate | 0.0001 |
| | Batch Size | 256 |
| Stage 2 | N layers | 6 |
| | N heads | 6 |
| | Embedding Dim. | 384 |
| | Attn. Dropout | 0.1 |
| | Embedding Dropout | 0.1 |
| | Beam size | 5 |
| | Temperature | 1.0 |
| | Learning Rate | 0.0001 |
| | Batch Size | 128 |
| Stage 3 | Base model | resnet18 |
| | Learning Rate | 0.0001 |
| | Batch Size | 128 |

## B.2 IMPLEMENTATION DETAILS OF BASELINES

**AVDC.** We follow the codebase[1] to train the video model. To fit within the memory constraints of a single 24GB GPU, we reduce the batch size to 2. For converting generated videos into executable actions, we adopt the approach provided in the official codebase[2], which utilizes optical flow, depth, and segmentation masks. We evaluate the results only on MetaWorld, as we were unable to get the action transformation pipeline to work in the LIBERO environment due to the need for additional environment-specific design. Recent work (Luo & Du, 2025) also reports near-zero success rates on LIBERO.

**LAPA.** We follow the official codebase[3] to train the model on MetaWorld and LIBERO. During the latent pretraining stage, we reduce the number of training steps to 1,000, given the small-scale datasets with only 50 demonstrations per task. In the action fine-tuning stage, we fine-tune the model for 500 steps. All experiments are conducted using 8 V100 GPUs.

## B.3 CROSS-EMBODIMENT TASKS

The task sets used in setting (**b**) of cross-embodiment transfer are listed in Table 8.

Table 8: **Cross-embodiment transfer task assignment.**

| Task set | | | | | |
|---|---|---|---|---|---|
| **Set A** | door-open | door-close | basketball | hammer | button-press-topdown |
| **Set B** | faucet-close | faucet-open | handle-press | button-press | assembly |

---

[1] https://github.com/flow-diffusion/AVDC
[2] https://github.com/flow-diffusion/AVDC_experiments/tree/main
[3] https://github.com/LatentActionPretraining/LAPA

## B.4 LATENCY ANALYSIS

We measure end to end control frequency on a single RTX 4090. The SOF pipeline runs at about 15 Hz from vision input to predicted skill tokens to decoded flow to actions. Under the same setup, Diffusion Policy runs at about 10 Hz.

## C  COMPUTATIONAL RESOURCES

We use the workstations listed in Table 9. Our method requires approximately 20 hours for the first stage, 3 hours for the second stage, and 3 hours for the third stage, totaling around 26 GPU hours on a single workstation. For reference, the training cost of comparable baselines is on a similar scale: behavior cloning (BC) takes about 1 hour for 30 demonstrations, Diffusion Policy requires roughly 4 hours for 30 demonstrations, AVDC uses approximately 24 GPU hours in total (12 hours for video diffusion training on 2 RTX 4090 GPUs plus a training-free Flow2Action module), and LAPA requires about 30 GPU hours (3 hours for latent action quantization on 2 V100s, 2 hours for latent pretraining on 8 V100s, and 1 hour for finetuning on 8 V100s). Overall, the computational cost of our method is comparable to or slightly lower than recent action-free video pretraining approaches, while delivering consistent improvements across tasks and benchmarks.

Table 9: **Computational resources.**

| Workstation | CPU | GPU | RAM |
|---|---|---|---|
| Workstation 1 | Intel Xeon w7-2475X | NVIDIA GeForce RTX 4090 x 2 | 125 GiB |
| Workstation 2 | Intel Xeon w5-2455X | NVIDIA RTX A6000 x 2 | 125 GiB |
| Workstation 3 | Intel Xeon W-2255 | NVIDIA GeForce RTX 4070 Ti x 2 | 125 GiB |

## D  LLM USAGE

We used LLMs solely for language polishing. All research ideas and related work were developed independently, without reliance on LLMs.

