# OpenReview forum: "Learning Skills from Action-Free Videos"
_ICLR.cc/2026/Conference — ICLR 2026 Conference Withdrawn Submission_

### Official Review · Reviewer_F926 · 2025-10-29

**Soundness:** 2
**Presentation:** 3
**Contribution:** 2
**Rating:** 4
**Confidence:** 5

**Summary:**

This paper introduces a method called SOF, which proposes a two-stage pipeline for learning policies from action-free video data and subsequently fine-tuning on a small set of action-labeled data. In the first stage, SOF utilizes optical flow as the modality for learning "skill tokens," contrary to the commonly used RGB modality. A Transformer decoder is then used to auto-regressively predict these skill tokens. The second stage involves fine-tuning a "translator" to map the learned skill tokens to concrete robot actions.

**Strengths:**

- The paper is easy to follow, and the figures are clear and intuitive, significantly aiding in the understanding of the proposed architecture.
- The core motivation is well-defined. The choice to use optical flow instead of RGB frames for skill token learning is sensible, as it inherently removes redundant background and static scene information, thereby potentially increasing the information density of the skill tokens.

**Weaknesses:**

Limited and Simplified Experimental Setup: The experimental validation is too simplistic and limited in scope.

- The paper only tests on 9 Meta-World tasks (out of a total of 50).

- For the Libero benchmark, only 10 Libero-Goal and 10 Libero-Long tasks are evaluated. Standard practice for Libero typically involves training and testing across all 4 Libero task suites (totaling 40 tasks).

- While BridgeData is mentioned, it is only utilized for visualization analysis and not for policy training or core performance evaluation.

Overall, the simulated tasks tested are too few and overly simple, and there is a lack of real-world robot experiments.

**Questions:**

- Baseline Comparison with LAPA: Since BC and DP cannot leverage action-free datasets, their poor performance is expected. Given that AVDC is only deployable on Meta-World, LAPA stands out as the most relevant and crucial baseline for SOF. However, the results show that LAPA achieves an all-zero score on Libero-Goal and is missing entirely on Libero-Long. This is highly peculiar, as LAPA generally shows decent performance on both the SIMPLER-Bridgedata and real-world tasks, and an all-zero score on Libero seems fundamentally unlikely. Could the authors clarify the cause of this extremely poor performance? Is it potentially due to unsuitable training parameters or code bugs? Furthermore, I strongly recommend that LAPA not be omitted from the results on Libero-Long.

- Clarification on Stages and Data Usage (Section 5.5): In Section 5.5, I would appreciate it if the authors could explicitly clarify what "stage 1" and "stage 2" specifically refer to. Based on Figure 2, "stage 1" might refer to "Learning Skill Abstraction and Policy Learning on Skill Space," and "stage 2" might be "Flow to Action." However, the Appendix suggests "stage 1" might only denote "Learning Skill Abstraction."Additionally, how is the dataset utilized during the "Flow to Action" stage? Was data from only a single embodiment or both used?

- Validation of Transfer Experiments: The analysis experiments regarding task and embodiment transfer in Section 5.5 are very interesting. However, Meta-World tasks are quite simple. In the standard Meta-World task suite, the action space is only 4-dimensional ($x, y, z, \text{gripper}$), which largely avoids complex issues related to different robot embodiments/configurations. Could the authors validate these two points (task and embodiment transfer) in real-robot experiments to demonstrate the method's robustness in more complex, high-dimensional settings?

---

### Official Review · Reviewer_VD5v · 2025-10-30

**Soundness:** 2
**Presentation:** 3
**Contribution:** 2
**Rating:** 2
**Confidence:** 5

**Summary:**

This paper introduces SOF, a framework that learns reusable robotic skills directly from action-free videos by leveraging optical flow as a surrogate for actions. The method extracts discrete skill tokens from flow sequences, plans in this skill space, and translates the plan into executable actions. Experiments demonstrate that SOF improves performance in multi-task and long-horizon settings and enables skill transfer across different robot embodiments, offering a more scalable approach to learning from unstructured visual data.

**Strengths:**

The idea is well motivated, and the writing is easy to follow. The authors conduct experiments on three different domains: MetaWorld, LIBERO, and BridgeData.

**Weaknesses:**

1.	The core components of the method, which leverage optical flow as a mid-level representation and learns discrete skill tokens, are well-established in recent literature[1,2]. The paper's primary contribution appears to be the combination of these two ideas into a single framework, but it provides limited new algorithmic insight or theoretical foundation beyond this synthesis.
2.	The paper positions only using a third-person view as an advantage over other methods like ATM (using a wrist camera) or QueST (using proprioception). However, this critique is misaligned with practical robotics, where wrist cameras and proprioceptive states are standard and crucial for achieving robust, fine-grained, and contact-rich manipulation. The authors' chosen setup may inherently limit their method's applicability to simpler, pick-and-place-style tasks.
3.	The absence of a direct comparison with the highly relevant UniVLA baseline is a significant omission. Given that UniVLA follows a conceptually similar latent action pretraining approach (but in image space) and has demonstrated superior performance (e.g., >90% on LIBERO), the presented results for SOF (25% on LIBERO-GOAL) are underwhelming and raise questions about the relative effectiveness of the optical-flow-based approach.
4.	The policy learning experiments are only conducted in simulation (MetaWorld and LIBERO) without any real-world robot experiments, making it difficult to evaluate the method's practicality and robustness. Furthermore, the decision to evaluate only 4 out of 10 tasks in the standard LIBERO-10 benchmark, without a clear justification, reduces the rigor and comparability of the long-horizon claims.

Therefore, I think this paper has not achieved the acceptance threshold of ICLR.

[1] Ko P C, Mao J, Du Y, et al. Learning to act from actionless videos through dense correspondences

[2] Bu Q, Yang Y, Cai J, et al. Univla: Learning to act anywhere with task-centric latent actions

**Questions:**

Please refer to the Weaknesses part.

---

### Official Review · Reviewer_xkUP · 2025-10-31

**Soundness:** 2
**Presentation:** 4
**Contribution:** 2
**Rating:** 4
**Confidence:** 5

**Summary:**

This paper presents a method for learning robotic skills through quantized flow representation prediction. The approach’s action-free property enables cross-embodiment skill transfer. Experimental results demonstrate that the proposed quantized flow representation outperforms both video-flow pipelines (e.g., AVDC) and implicit quantized representations (e.g., LAPA).
My main concern is that the proposed method appears to be a straightforward combination of several existing techniques (2D flow + quantization + action-free formulation), which limits its novelty and overall contribution. Nevertheless, validating a new form of predictable representation is still a meaningful contribution. Another issue is the absence of comparison with key baselines, particularly direct flow prediction methods (e.g., ATM).

**Strengths:**

1. The paper proposes quantizing flow representations to enhance predictability for Transformer-based architectures. Experiments show that this strategy effectively supports action-free skill learning.
2. The writing is clear and well-organized, with high-quality visualizations.
3. The analysis of skill tokens and progress provides interesting insights.

**Weaknesses:**

1. A major weakness is the lack of comparison with the most critical baseline, i.e., direct flow prediction methods (e.g., ATM). Since the primary claimed contribution lies in introducing quantized flow representation, omitting this comparison undermines the soundness of the work.
2. The novelty and contribution are limited, as the approach appears to be a direct integration of prior works (e.g., LAPA quantization, ATM flow, and AVDC flow-to-action frameworks). While validating quantized flow representations is valuable, the overall conceptual contribution may not meet the ICLR-level threshold.

**Questions:**

1. Please include a comparison with direct flow prediction baselines (e.g., ATM). Without this, the evaluation feels incomplete and may negatively impact the overall score after the rebuttal.
2. Please clarify the motivation and conceptual novelty of the proposed approach, particularly in relation to the combination of existing methods.

---

### Note · Authors · 2025-11-22

**Comment:**

We thank the reviewers and the AC for their careful evaluation. We recognize that the current version of the manuscript does not clearly convey its novelty and the experiments are not comprehensive enough. Therefore, we have decided to withdraw the submission to allow for substantial revisions.

**Withdrawal Confirmation:**

I have read and agree with the venue's withdrawal policy on behalf of myself and my co-authors.